# Deubiquitinating Enzyme USP8 Is Essential for Skeletogenesis by Regulating Wnt Signaling

**DOI:** 10.3390/ijms221910289

**Published:** 2021-09-24

**Authors:** Sachin Chaugule, Jung-Min Kim, Yeon-Suk Yang, Klaus-Peter Knobeloch, Xi He, Jae-Hyuck Shim

**Affiliations:** 1Shim Lab, Division of Rheumatology, Department of Medicine, University of Massachusetts Chan Medical School, Worcester, MA 01605, USA; sachin.chaugule@umassmed.edu (S.C.); jungmin.kim@umassmed.edu (J.-M.K.); yen.yang@umassmed.edu (Y.-S.Y.); 2Institute of Neuropathology, Medical Faculty, University of Freiburg, 79106 Freiburg, Germany; klaus-peter.knobeloch@uniklinik-freiburg.de; 3F. M. Kirby Neurobiology Center, Boston Children’s Hospital, Department of Neurology, Harvard Medical School, Boston, MA 02115, USA; xi.he@childrens.harvard.edu; 4Horae Gene Therapy Center, University of Massachusetts Chan Medical School, Worcester, MA 01605, USA; 5Li Weibo Institute for Rare Diseases Research, University of Massachusetts Chan Medical School, Worcester, MA 01605, USA

**Keywords:** USP8, deubiquitinating enzyme, osteoblast, FZD5, Wnt signaling, skeletogenesis

## Abstract

Disturbance in a differentiation program of skeletal stem cells leads to indecorous skeletogenesis. Growing evidence suggests that a fine-tuning of ubiquitin-mediated protein degradation is crucial for skeletal stem cells to maintain their stemness and osteogenic potential. Here, we demonstrate that the deubiquitinating enzyme (DUB) ubiquitin-specific protease 8 (USP8) stabilizes the Wnt receptor frizzled 5 (FZD5) by preventing its lysosomal degradation. This pathway is essential for Wnt/β-catenin signaling and the differentiation of osteoprogenitors to mature osteoblasts. Accordingly, deletion of USP8 in osteoprogenitors (*Usp8^Osx^*) resulted in a near-complete blockade in skeletal mineralization, similar to that seen in mice with defective Wnt/β-catenin signaling. Likewise, transplanting USP8-deficient osteoprogenitors under the renal capsule in wild-type secondary hosts did not to induce bone formation. Collectively, this study unveils an essential role for the DUB USP8 in Wnt/β-catenin signaling in osteoprogenitors and osteogenesis during skeletal development.

## 1. Introduction

Skeletal stem cells (SSCs) are pluripotent cells that can self-renew and differentiate into osteoblasts, chondrocytes, adipocytes, and stromal cells [1]. SSCs are committed to osteoprogenitors, which subsequently proliferate, condense, and differentiate into mature osteoblasts [2]. Osteoblast differentiation is predominantly governed by master transcriptional factors, such as runt-related transcription factor 2 (*Runx2*) and osterix (*Osx*) [3,4], and by key signaling pathways such as Wnt and Notch signaling [5,6], whose dysregulation leads to skeletal abnormalities. Recent studies indicate that ubiquitin-mediated protein degradation has a critical role in skeletal formation [7].

The ubiquitin-mediated proteasomal degradation pathway is essential for controlling various cellular processes, such as protein stability, protein interactions, intracellular transport, and transcriptional activity [8]. This pathway is tightly regulated by the balance between enzymatic activities of ubiquitin E3 ligases and deubiquitinating enzymes (DUBs), which play a critical role in maintaining the stemness and osteogenic potential of SSCs [7,9]. Based on sequence predictions, the human genome contains approximately 100 putative DUBs that include the subclasses of ubiquitin C-terminal hydrolases, ubiquitin-specific proteases (USPs), Machado–Joseph disease protein domain proteases, ovarian tumor proteases, JAB1/MPN/Mov34 metalloenzyme (JAMM) motif proteases and MINDY DUBs [10,11,12]. Previous studies have suggested the DUBs, USP1 and USP4 preserve mesenchymal stem cell programs and antagonize osteoblast differentiation, respectively [13,14]. The DUB USP8 (UBPy/HUMORF8) was originally identified as a growth regulated DUB [15]. USP8 also functions as a key regulator of endosomal sorting and trafficking of cargo proteins in yeast [16] and also participates in regulating the growth of various cancer cells [17]. Germline deletion of USP8 in mice causes early embryonic lethality due to growth inhibition and reduced expression of multiple receptor tyrosine kinases (RTKs), including epidermal growth factor receptor (EGFR), mesenchymal-epithelial transition factor (c-MET), erbB-2 receptor (ERBB2), and ERBB3 [16,18]. Additionally, USP8 has been reported to deubiquitinate EPG5, which is important for autophagic regulation in embryonic stem cells and maintenance of their stemness [19]. Finally, two recent studies demonstrated that USP8 controls Wnt signaling during *Xenopus* and *Drosophila* development by stabilizing the frizzled receptor (FZD) [20,21]. Using CRISPR/Cas9 genome-wide screening in human cells, we have previously identified TMEM79/MATTRIN, an orphan multi-span transmembrane protein, as an inhibitor of Wnt signaling. TMEM79 inhibits USP8-mediated deubiquitination of FZD, promoting FZD degradation [20]. However, the importance of this pathway in the skeleton remains largely unknown. Here, we demonstrate that conditional deletion of USP8 in osteoprogenitors impaired osteogenesis and bone formation. Mechanistically, USP8 prevents the degradation of ubiquitinated FZD and secures Wnt-induced osteogenesis. This study establishes a USP8-FZD pathway controlling Wnt/β-catenin signaling as a fine-modulator of bone formation during skeletal development.

## 2. Results

### 2.1. USP8 Controls Wnt/β-Catenin Signaling in Osteoblasts

To explore the role of USP8 in skeletogenesis, we examined the expression of USP8 at an early stage of postnatal skeletal development (P0). Immunohistochemistry (IHC) demonstrated that USP8 is highly expressed in osteoblasts and osteocytes (Figure 1A). Likewise, in vitro osteogenic culture of mouse skeletal stem cells (SSCs) showed that USP8 expression gradually increased during osteoblast differentiation (Figure 1B,C). Next, we conditionally deleted *Usp8* in osteoprogenitors by crossing mice containing a *Usp8* (*Usp8^fl/fl^*)-floxed allele [16,22] with the Osterix-Cre deleter strain to generate *Usp8^Osx^* mice [23]. Primary osteoblast precursors (COB) were isolated form the calvaria of *Usp8^fl/fl^* and *Usp8^Osx^* neonates at the age of postnatal day 0, and USP8 expression was assessed by immunoblotting analysis, indicating efficient deletion of *Usp8* in *Usp8^Osx^* COBs (Figure 1D).

Given our previous study showing that USP8 controls Wnt/β-catenin signaling by deubiquitinating the frizzled receptor 5 (FZD5) for stabilization [20], we hypothesized that USP8 deficiency may decrease the protein stability of FZD5 and downregulate Wnt/β-catenin signaling in osteoblast precursors. Protein levels of β-catenin were markedly decreased in *Usp8^Osx^* COBs relative to *Usp8^fl/fl^* COBs, which was even further reduced in mature osteoblasts (Figure 1D). This result is consistent with an IHC analysis showing a significant decrease in protein levels of β-catenin in the primary ossification center of E18.5 *Usp8^Osx^* femurs (Figure 1E). Unlike *Usp8^fl/fl^* femurs, *Usp8^Osx^* femurs showed the lack of skeletal elements in the primary ossification center.

### 2.2. USP8 Stabilizes the Frizzled Receptor in Osteoblasts

Our previous study identified USP8 as the deubiquitinating enzyme (DUB) of FZD5. USP8 interacts with FZD5, and the DUB domain of USP8 was sufficient to inhibit the ubiquitination of FZD5 while enhancing Wnt/β-catenin signaling [20]. To further investigate USP8-mediated stabilization of FZD5 in osteoblast precursors, co-immunoprecipitation analysis was performed in wild type COBs, confirming endogenous interaction between USP8 and FZD5 (Figure 2A). USP8-deficiency downregulated protein levels of FZD5 in mature osteoblasts (Figure 2B). Immunoblotting analysis revealed that treatment with the lysosomal inhibitor Bafilomycin A1 (Baf A1) was effective to suppress immature FZD degradation in *Usp8*-deficient COBs while having little to no effect on mature FZD5 expression (Figure 2C). Previous reports have shown that the treatment with Baf A1 preferentially inhibits lysosomal degradation of immature FZD receptors that are primarily localized in the endoplasmic reticulum (ER) relative to mature FZD receptors located in the plasma membrane [24,25]. In this regard, USP8, predominantly localized in intracellular membranes, is more effective for the stabilization of the immature form of FZD5 than the mature form [20]. Notably, ubiquitinated FZD5 was markedly accumulated in the absence of USP8 when its lysosomal degradation was blocked by Baf A1 treatment (Figure 2D). Thus, these results suggest that USP8-mediated deubiquitination controls the protein stability of FZD5 in osteoblast precursors by suppressing lysosomal degradation of ubiquitinated FZD5.

To determine which domains of USP8 are essential for regulation of Wnt/β-catenin signaling, *Usp8*-deficient COBs were transfected with plasmids to express the C-terminal (USP8C) or N-terminal region (USP8N) of USP8, and Top-Flash luciferase activity and immunoblotting were analyzed for β-catenin activity and expression, respectively (Figure 2E–G). The reduced expression and activity of β-catenin in *Usp8*-deficient COBs were partly reverted by enforced expression of USP8C containing the deubiquitination enzyme domain and the 14-3-3 binding motif. In contrast, little effect was observed upon overexpression of USP8N containing the MIT and Rhod domains. This result suggests that the catalytic domain of USP8 is sufficient to restore Wnt/β-catenin signaling in the absence of USP8. Moreover, enforced expression of FZD also reverted the expression and activity of β-catenin in *Usp8*-deficient COBs (Figure 2E–G), suggesting that FZD5 functions downstream of USP8 in osteoblast precursors. Thus, these results demonstrate that the DUB USP8 is required for Wnt/FZD/β-catenin signaling in osteoblast precursors.

### 2.3. USP8 Deletion in Osteoprogenitors Impairs Bone Formation during Skeletogenesis

Since germline deletion of USP8 results in early embryonic lethality in mice [16], we had previously generated mice in which USP8 was conditionally deleted in the limb-specific mesenchyme by crossing mice containing a *Usp8* (*Usp8^fl/fl^*)-floxed mice with a Prx1-Cre mouse line to generate *Usp8^Prx1^*mice [26]. However, these mice also die around embryonic day 9 as Prx1-driven expression of Cre recombinase occurs during embryonic development [27]. Thus, we next examined skeletal phenotypes of *Usp8^Osx^* mice to investigate the role of USP8 in osteoprogenitors. Severe skeletal abnormalities were observed in E18.5 and P0 *Usp8^Osx^* pups. P0 *Usp8^Osx^* pups died shortly after birth due to respiratory distress (Appendix A). Alizarin red and alcian blue staining of skeletal preparations revealed that ossification (red) was markedly reduced in the calvaria, scapula, humerus, radius, ulna, femur, tibia, fibula, and ribs of *Usp8^Osx^* pups and the clavicles of *Usp8^Osx^* pups were hypoplastic (Figure 3A,B, and Appendix A). Of note, cartilage (blue) is normally formed in the skeleton of E15.5 *Usp8^Osx^* embryos (Figure 3C and Appendix A), suggesting that USP8 function in osteoprogenitors is dispensable for cartilage development. Additionally, endochondral ossification of long bones was almost completely arrested at the earliest stages of primary ossification center formation, where little to no mature osteoblasts expressing osteocalcin was detected (Figure 3D,E and Appendix A). These skeletal phenotypes are similar to those seen in mice lacking β-catenin in osteogenic progenitors [28,29], suggesting that USP8 is essential for skeletal development by regulating Wnt/β-catenin signaling in osteoprogenitors.

### 2.4. USP8 Is Required for Osteogenic Differentiation of Skeletal Progenitors

Skeletogenesis occurs through a developmental hierarchy of bone lineage-specific skeletal progenitors. Thus, we examined the percentage of different subpopulations of skeletal progenitors. Subpopulations of mouse skeletal progenitors, including skeletal stem cells (SSCs); pre-bone cartilage, and stromal progenitors (pre-BCSP); and bone, cartilage, and stromal progenitors (BCSP), can be sorted using fluorescent-activated cell sorting (FACS) from the limbs of E18.5 *Usp8^fl/fl^* and *Usp8^Osx^* embryos based on the expression of cell surface markers [30] (Appendix A). The frequency of these skeletal progenitors was comparable between *Usp8^fl/fl^* and *Usp8^Osx^* embryonic limbs (Figure 4A), demonstrating that USP8 function is dispensable for the maintenance of skeletal progenitors. Immunoblotting analysis of FACS-sorted *Usp8^Osx^* SSCs confirmed efficient deletion of USP8 (Figure 4B). Protein levels of FZD5 and β-catenin were markedly decreased in the absence of USP8 during osteoblast differentiation (Figure 4B). Unlike human USP8-deficient cells showing abnormal cell cycling arrest [31], cell proliferation rate was comparable between FACS-sorted *USP8^fl/fl^* and *Usp8^Osx^* SSCs (Figure 4C), suggesting that USP8 is dispensable for cell-cycle regulation. Similarly, USP8-deficiency in embryonic stem cells did not affect cell viability and death [19]. Of note, the vitro osteogenic potential of these *Usp8^Osx^* SSCs was markedly reduced as shown by the significant reduction in markers for early osteoblast differentiation, alkaline phosphatase (ALP) activity, and osteopontin (*Opn*) expression (Figure 4D,E). This result is consistent with our findings that *Usp8*-deficiency in COBs results in a significant decrease in protein levels of FZD5 and β-catenin (Figure 1D and Figure 2B) as well as in compromised ALP and mineralization activity (Appendix A). Altogether, our findings demonstrate an essential role for USP8 in osteoblast differentiation of SSCs by controlling Wnt/β-catenin signaling.

To determine their osteogenic potential in vivo, SSCs were sorted by FACS from E18.5 *Usp8^fl/fl^* and *Usp8^Osx^* embryonic limbs, transplanted into the kidney capsule of wild-type secondary hosts, and bone formation was assessed by microCT and histology. Bone formation normally occurs in the kidney capsule when transplanted with *Usp8^fl/fl^* SSCs. However, despite engrafting within the kidney capsule, little to no bone mass was detected in the kidney capsule transplanted with *Usp8^Osx^* SSCs (Figure 4F,G). These data demonstrate an impaired osteogenic potential of *Usp8^Osx^* SSCs in vivo. Taken together, as schematically shown in Figure 4H, USP8-mediated stabilization of FZD5 is important for the Wnt/β-catenin signaling, osteogenesis, and skeletal development.

## 3. Discussion

In addition to our previous study identifying USP8 as an FZD5-stabilizing DUB in Wnt signaling [20], this study further demonstrated that the DUB USP8 is essential for Wnt/β-catenin signaling in osteoblast precursors, osteogenesis, and bone formation during skeletal development. Using the HEK293T overexpression system, we previously demonstrated that USP8 interacts with FZD5 and the DUB domain of USP8 inhibits ubiquitination of FZD5 while enhancing Wnt signaling. Here, we demonstrated that in osteoblast precursors, USP8 interacts with FZD5, and its deficiency increases ubiquitination of FZD5, leading to a decrease in protein levels of FZD5 and β-catenin and thereby Wnt/β-catenin signaling activity. Notably, USP8 stabilizes FZD5 by preventing lysosomal degradation of ubiquitinated FZD5. Additionally, USP8 deletion in osteoprogenitors (*Usp8*^Osx^) downregulates protein levels of FZD5 and β-catenin. These mice phenocopied the skeletal phenotypes seen in mice lacking β-catenin in osteogenic progenitors [27,30]. Collectively, the USP8–FZD5 signaling axis plays a critical role in the regulation of Wnt/β-catenin signaling in osteoblast precursors, required for osteogenesis and bone formation during skeletal development (Figure 4H).

Given that USP8 does not interact directly with β-catenin and alter its ubiquitination (data not shown), lack of FZD expression is likely to reduce both expression and transcription activity of β-catenin, resulting in a defect of Wnt-induced osteogenesis and skeletogenesis. Previous studies using yeast mutagenesis have reported that Doa4, the yeast ortholog of USP8, deubiquitinates ubiquitinated cargo proteins upon endosomal sorting and trafficking to maintain a free ubiquitin pool [32]. Likewise, the mammalian DUB USP8 has been reported to deubiquitinate various ubiquitinated G-protein coupled receptors, including smoothened (SMO), and FZD, and RTKs such as EGFR for stabilization, which are involved in the signaling pathways of Hedgehog, Wnt, EGF, and other growth factors [16,21,33,34]. Additionally, USP8 plays a role in maintaining the stemness of embryonic stem cells via deubiquitinating the eukaryotic autophagic regulator, EPG5 [19]. Thus, we note that while we demonstrated FZD5 as a critical substrate for USP8 in osteoprogenitors, USP8 is likely to have additional substrates that contribute to the regulation of osteogenesis. Taken together, the USP8-mediated stabilization of FZD5 is a key molecular pathway that controls bone formation during skeletal development via regulation of Wnt/β-catenin-mediated osteogenesis.

## 4. Materials and Methods

### 4.1. Animals

*Usp8^fl/fl^* mice were generated as previously described [16,22] and were maintained in the C57BL/6 background. To generate osteoprogenitor-specific knockouts, *Usp8^fl/fl^* mice were crossed with Osx-Cre, both male and female *Usp8^Osx^* mice, and littermate controls were randomized and used for all skeletal analyses in a blind manner without any excluded mice. All animals, including USP8-floxed and knockout mice, were used under the NIH Guide for the Care and Use of Laboratory Animals and were handled according to the animal protocol approved by the University of Massachusetts Chan Medical School Institutional Animal Care and Use Committee (IACUC) [IACUC, D16-00196 (A-3306-01), 4 January 2017].

### 4.2. Skeletal Preparation, Histology, and Immunohistochemistry

For skeletal preparation, postnatal day 0 (P0), embryonic day 18.5 (18.5), and E15.5 pups were skinned, eviscerated, fixed in 95% EtOH for 48 h, and the skeletons were transferred to acetone for 48 h. Further, skeletons were stained with 0.1% Alizarin red and 0.3% Alcian blue solution (Sigma-Aldrich, A3157, St. Louis, MO, USA) for 3–4 days, as previously described [35]. After staining, samples were washed with 95% EtOH, and soft tissue was cleared in 1% KOH solution. All images presented are representative of the respective genotypes (*n* ≥ 3).

For histological analysis, long bones from P0, E15.5 and E18.5 *Usp8^fl/fl^* and *Usp8*^Osx^ mice were fixed in 10% neutral formalin buffer at 4 °C for 2 days and then decalcified in 15% tetrasodium EDTA buffer (pH 8.0) at 4 °C for 24 h. Tissues were dehydrated in varying concentrations of EtOH, kept in xylene, and embedded in paraffin. Paraffin sections were performed at a 7 µm thickness along with the coronal plate and stained with Safranin O solution.

For immunohistochemistry, longitudinal sections of femurs of P0 and E18.5 *Usp8^fl/fl^* and *Usp8^Osx^* mice were stained with anti-USP8 (Gentex, GTX103747, Irvine, CA, USA), anti-BGLAP (Bioss ANTIBODIES, bs-0470R, Woburn, MA, USA), and anti-β-catenin antibodies (Cell Signaling, 9587, Danvers, MA, USA). Briefly, after deparaffinization and rehydration, paraffin tissue sections were incubated with a blocking solution (3% goat serum, 1% BSA, 0.1% Triton X-100 in Phosphate-Buffered Saline (PBS)) for 1 h at room temperature, incubated with primary antibody overnight at 4 °C, followed by TSA-biotin (Perkin Elmer, Waltham, MA, USA) and streptavidin-HRP (Santa Cruz Biotech, sc-2357 & sc-516102, Dallas, TX, USA), and then visualized with 3,3′ diaminobenzidine tetrahydrochloride as per the manufacturer’s instructions.

### 4.3. Cell Culture

To delete USP8 in pre-osteoblasts (∆USP8), primary calvarial osteoblasts (COBs) were isolated from calvaria of *Usp8^fl/fl^* neonates at postnatal day 0 using a digesting solution containing collagenase type II (5 mg/mL, Worthington, LS004176, Lakewood, NJ, USA) and dispase II (10 mg/mL, Roche, 4942078001, Mannheim, Germany). They were transduced with lentiviruses expressing Cre- recombinase, and the transduced cells were selected by puromycin. COBs were maintained in α-MEM medium (Gibco, A10490-01, Carlsbad, CA, USA) containing 10% FBS (Corning, 35-010-CV, Woodland, CA, USA), 2 mM L-glutamine (Thermo Fisher Scientific, 25030081, USA), 1% penicillin/streptomycin (Corning, 30-002-Cl, Manassas, VA, USA), and 1% nonessential amino acids (Corning, Manassas, VA, USA).

### 4.4. Osteoblast Differentiation Assay

For osteoblast differentiation, primary COBs were cultured in an osteogenic medium containing 0.1 mg/mL ascorbic acid (Sigma-Aldrich, A8960, Saint Louis, MO, USA), and 10 mM β-glycerophosphate (Sigma, G9422, USA), and incubated with Alamar blue solution (Invitrogen, DAL1100, Carlsbad, CA, USA; Dilution 1:10) for cell proliferation. Subsequently, cells were washed and incubated with a solution containing 6.5 mM Na_2_CO_3_, 18.5 mM NaHCO_3_, 2 mM MgCl_2_, and phosphatase substrate (Sigma-Aldrich, S0942 St. Louis, MO, USA), and alkaline phosphatase activity was measured by an iMark microplate absorbance reader (Bio-Rad, 1681130, Hercules, CA, USA). To assess extracellular matrix mineralization in mature osteoblasts, cells were washed twice with PBS and fixed in 4% paraformaldehyde for 20 min at room temperature. Fixed cells were washed twice with distilled water and stained with a 2% Alizarin red solution (Sigma-Aldrich, A5533, St. Louis, MO, USA) for 10–20 min. Cells were then washed three times with distilled water and examined for the presence of calcium deposits.

### 4.5. Quantitative Real Time-PCR Analysis

Total RNA was purified from cells using QIAzol lysis reagent (QIAGEN, 79306, Germantown, MD, USA) and cDNA was synthesized using the High-Capacity cDNA Reverse Transcription Kit from Applied Biosystems (Thermo Fisher Scientific, 4368814, Carlsbad, CA, USA). Quantitative RT-PCR was performed using iTaqTM Universal SYBR^®^ Green Supermix (Bio-Rad, 1725122, Hercules, CA, USA) with CFX connect RT-PCR detection system (Bio-Rad, Hercules, CA, USA) and normalized to β-actin. Primer sequences used for PCR are described in Appendix A.

### 4.6. Luciferase Reporter Assay

WT and ∆USP8 COBs were transfected with a β-catenin-responsive reporter gene (TopFlash-Luc), along with the Renilla luciferase vector (Promega, Madison, WI, USA) using the Effectene Transfection Reagent (Qiagen, 301427, Germantown, MD, USA). After transfection, a dual-luciferase assay was performed according to the manufacturer’s protocol (Promega, E1910, Madison, WI, USA), and luciferase activity was normalized to Renilla.

### 4.7. Western Blotting

Osteoblasts were lysed in NP40 lysis buffer (50 mM Tris, 50 mM NaCl, 1% Triton X-100, 1 mM EDTA, 10 mM NaF, proteinase inhibitor cocktails (Sigma, P8340)), subjected to SDS-PAGE, and then transferred to Immobilon-P membranes (Millipore, Burlington, MA, USA). After incubation with a blocking buffer (5% skim milk in TTBS buffer (100 mM Tris-HCl, pH 7.5, 150 mM NaCl, 0.1% Tween-20)), membranes were incubated with the indicated primary antibodies. Antibodies specific to USP8 (Gentex, GTX103747, Irvine, CA, USA), FZD5 (Abclonal, A12775, Woburn, MA, USA), β-catenin (Cell Signaling, 9587, Danvers, MA, USA), and GAPDH (Sigma-Aldrich, CB1001, St. Louis, MO, USA) were used according to the manufacturer’s instructions. Anti-GAPDH antibody was used as a loading control. Finally, washed blots were incubated with HRP-conjugated secondary antibodies, and developed with ECL (Thermo Fisher Scientific, Waltham, MA, USA).

### 4.8. Ubiquitination and Co-Immunoprecipitation Analyses

To examine endogenous interaction between USP8 and FZD5, mouse COBs were lysed with lysis buffer (50 mM Tris-HCl (pH 7.4), 150 mM NaCl, 1% Triton X-100, 1 mM EDTA, 1 mM EGTA, 50 mM NaF, 1 mM Na_3_VO_4_, 1 mM PMSF, and protease inhibitor cocktail (Sigma)) and immunoprecipitated with IgG control or anti-FZD5 antibodies along with protein G-conjugated agarose.

For ubiquitination analysis, WT and ΔUSP8 COBs were transfected with HA-FZD5, His-ubiquitin in the absence or presence of the indicated DNA constructs using the Effectene transfection reagent (Qiagen). Then, 24 h later, cells were treated with Bafilomycin A1 (10 nM, Sigma) overnight, lysed, and immunoprecipitated with HA-conjugated beads (Santa Cruz, sc-7392 AC, Dallas, TX, USA). The immunoprecipitated were subjected to SDS-PAGE, transferred to Immobilon-P membranes (Millipore), immunoblotted with the indicated antibodies, and developed with ECL (Thermo Fisher Scientific).

### 4.9. Isolation, Culture, and Kidney Transplantation of Skeletal Stem Cells (SSCs)

Limbs of E18.5 *Usp8^fl/fl^* and *Usp8^Osx^* embryos were dissociated by mechanical and enzymatic digestion using a digestion buffer (1 mg/mL of Collagenase P (Roche, 11213857, Mannheim, Germany), 2 mg/mL of Dispase II (Roche, 10165859001, Mannheim, Germany), 1 mg/mL of Hyaluronidase (Sigma-Aldrich, H3506, St. Louis, MO, USA), 10,000 units/mL of DNase I (Roche, 4716728001, Mannheim, Germany)) for 1 h at 37 °C, under gentle agitation. After digestion, cells were passed through a 40 μM cell strainer and washed with FACS buffer (2% fetal bovine serum in PBS (pH 7.2)). Cells were subsequently blocked with rat anti-mouse CD16/CD32 (BioLegend, 553141, San Diego, CA, USA) for 1 hour on ice, and stained with Biotin-conjugated CD45 (1:200, eBioscience, 13-0451-81, Grand Island, NY, USA), Tie2 (1:200, eBioscience, 13-5987-8, Grand Island, NY, USA) and Ter119 (1:200, eBioscience, 13-5921-8, Grand Island, NY, USA) with BV421-conjugated streptavidin (1:500, BioLegend, 405226, San Diego, CA, USA), PE-conjugated CD51 (1:200, eBioscience, 12-0512-81, Grand Island, NY, USA), BV605-conjugated Thy1.1–2 (1:200, BioLegend, 202537/140317, San Diego, CA, USA), APC-conjugated CD200 (1:100, BioLegend, 123809, San Diego, CA, USA) and PE/Cy7-conjugated CD105 (1:200, BioLegend, 120409, San Diego, CA, USA) for 45 min on ice. After washing three times, cells were resuspended in cold PBS (pH 7.2) containing 1 mM EDTA and 1 µg/mL Ghost Dye (Tonbo, 13-0865-T100, San Diego, CA, USA) and sorted using FACSAria II SORP cell sorter (Becton Dickinson, Franklin Lakes, NJ, USA) with the exclusion of dead cells and doublets. The gating strategy used for sorting skeletal progenitors is shown in Appendix A.

For kidney transplantation, the sorted skeletal progenitors (105 cells/mouse) were mixed with 20 μL of matrigel (Corning, 354230, Woodland, CA, USA) and injected underneath renal capsules of 8-week-old male mice (C57BL/6) after anesthetization. Eight weeks after transplantation, renal samples were fixed in a 4% PFA solution, and bone mass in the kidney was assessed by microCT. For microCT scanning, samples were scanned by microCT 35 (Scanco Medical, Southeastern, PA, USA) with a spatial resolution of 7 µM. 3D reconstruction images were obtained from contoured 2D images. For histology, samples were cryofixed, sectioned, and stained with Weigert Van Gieson staining solution. Sorted skeletal progenitors were subsequently cultured under osteogenic conditions as described earlier for primary COBs.

### 4.10. Statistical Analysis

All data are graphically represented as mean ± standard deviation and/or as box-and-whisker plots (with median and interquartile ranges) from min to max, with all data points shown unless otherwise noted. For experiments with three or more samples, statistical analysis was performed using a two-tailed, unpaired Student’s *t*-test. Values were considered statistically significant at *p* < 0.05. The results shown are representative of three or more distinct experiments. All statistical analyses were performed in GraphPad Prism software (GraphPad Software, Inc., La Jolla, CA, USA).

## Figures and Tables

**Figure 1 ijms-22-10289-f001:**
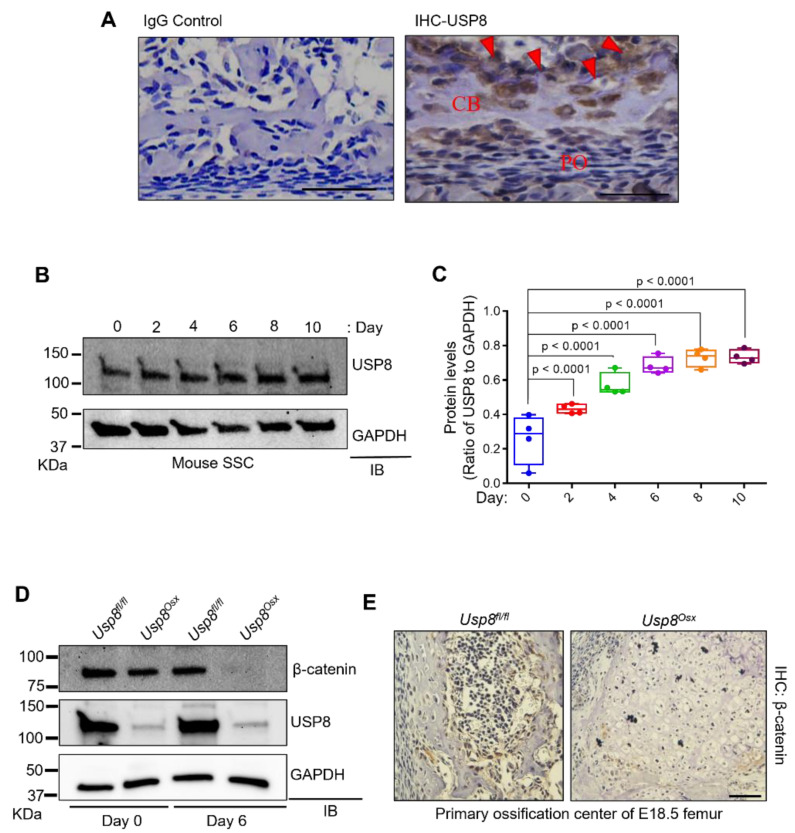
USP8 controls Wnt/β–catenin signaling in osteoblasts. (**A**) Immunohistochemistry (IHC) for USP8 in mouse femurs at postnatal day 0 (*n* = 3/group). Left, IgG control; right, IHC for USP8 in the cortical bone. CB, cortical bone; PO, periosteum. Arrows indicate bone-residing osteoblasts and osteocytes. Scale bar, 200 μm. (**B**,**C**) Kinetics of USP8 expression during osteoblast differentiation. Skeletal stem cells (SSCs, CD45^−^Ter119^−^Tie2^−^αVInt^+^Thy1^−^CD105^−^CD200^+^) isolated from E18.5 wild-type embryonic limbs were cultured under osteogenic condition, and USP8 expression was assessed by immunoblotting analysis. Protein levels of USP8 were displayed by the ratio of USP8 to GAPDH loading control (*n* = 4). (**D**) Primary osteoblast precursors (COBs) isolated from the calvaria of P0 *Usp8^fl/fl^* and *Usp8^Osx^* neonates (*n* = 5/group) were cultured under undifferentiated or osteogenic conditions for 6 days, and lysates were immunoblotted with the indicated antibodies. GAPDH was used as a loading control. (**E**) Immunohistochemistry for β–catenin in the primary ossification center of E18.5 *Usp8^fl/fl^* and *Usp8^Osx^* femurs (*n* = 3/group). Scale bar, 100 µm. Data are shown as a box-and-whisker plot (with median and interquartile ranges) from min to max, with all data points shown. Statistical analysis was performed using an unpaired Student’s t–test within Prism.

**Figure 2 ijms-22-10289-f002:**
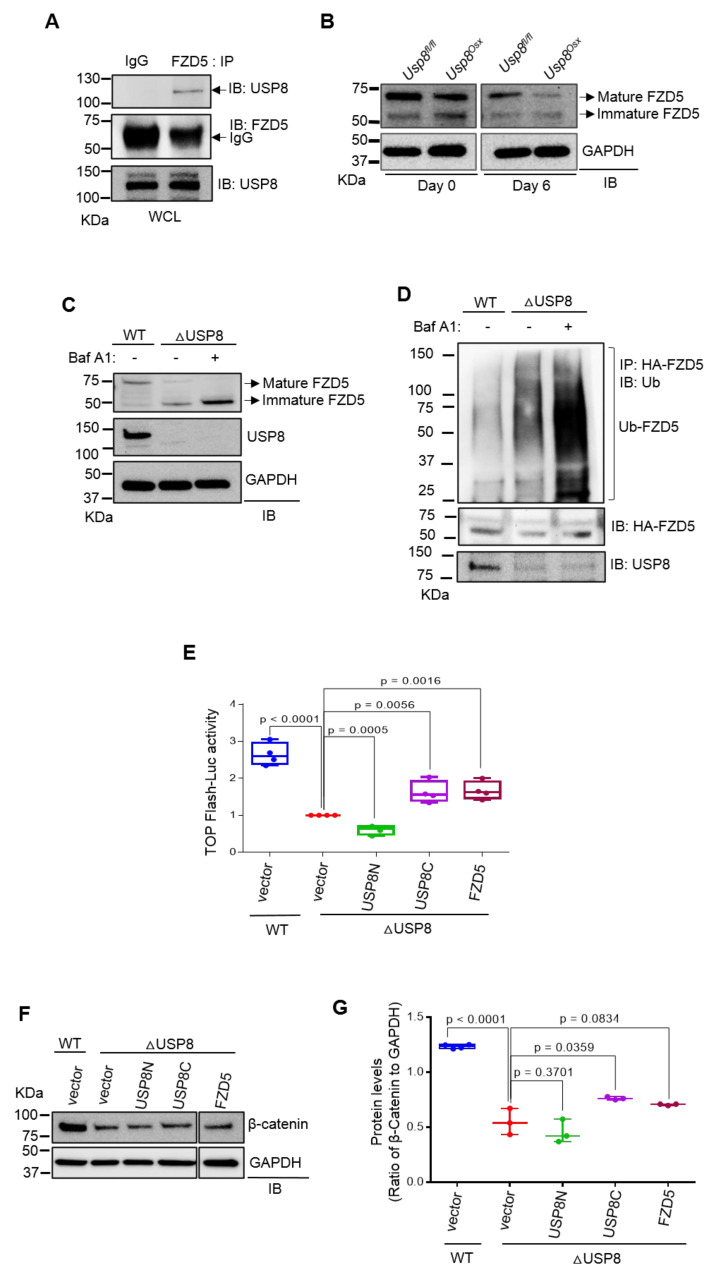
USP8 controls Wnt/FZD5/β–catenin signaling in osteoblasts. (**A**) Wild–type COBs were lysed, immunoprecipitated with anti–IgG control or anti–FZD5 antibody and protein G-conjugated agarose beads and immunoblotted with the indicated antibodies. WCL; whole cell lysate. (**B**) *Usp8^fl/fl^* and *Usp8^Osx^* COBs (*n* = 5/group) were cultured under undifferentiated or osteogenic conditions for 6 days, and lysates were immunoblotted with the indicated antibodies. GAPDH was used as a loading control. (**C**) *Usp8^fl/fl^* COBs were transduced with lentivirus expressing vector (WT) or CRE recombinase (ΔUSP8). Two days after puromycin–selection, cells were treated with vehicle or Bafilomycin A1 (Baf A1, 10 nM) overnight and immunoblotted with indicated antibodies. (**D**) HA–FZD5–expressing WT and ΔUSP8 COBs were treated with 10 nM of Baf A1 overnight, immunoprecipitated with anti–HA conjugated agarose, and immunoblotted with anti–ubiquitin antibody. (**E**) WT or ΔUSP8 COBs were transfected with a TopFlash luciferase reporter gene along with vector, USP8C, USP8N, or FZD5. Cells were cultured under osteogenic condition for 3 days, and β–catenin activity was assessed by measuring luciferase activity. (**F**,**G**) WT or ΔUSP8 COBs expressing vector, USP8C, USP8N, or FZD5 were immunoblotted with indicated antibodies. Protein levels of β–catenin were displayed by the ratio of β–catenin to GAPDH loading control (*n* = 3). Data are shown as a box-and-whisker plot (with median and interquartile ranges) from min to max, with all data points shown. Statistical analysis was performed using an unpaired Student’s t–test within Prism.

**Figure 3 ijms-22-10289-f003:**
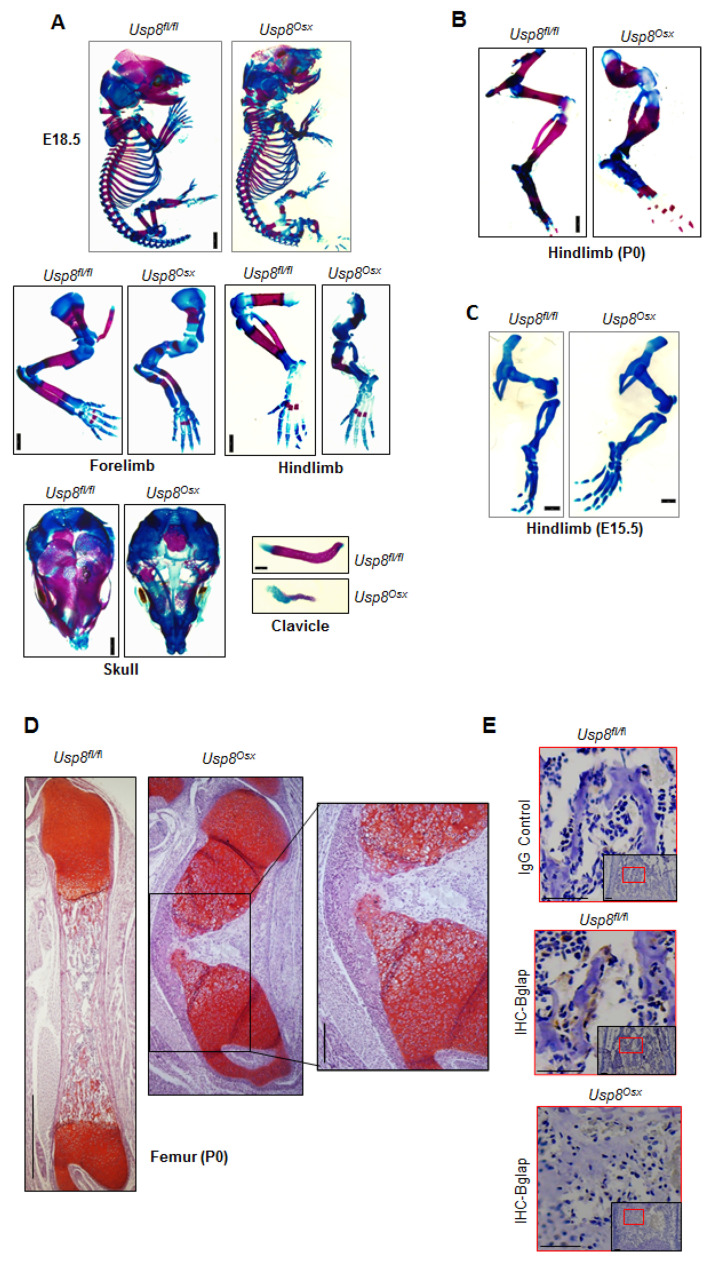
*Usp8^Osx^* mice display impaired ossification during skeletogenesis. (**A**) Alizarin red/alcian blue staining of skeletal preparations of whole body, forelimbs, hindlimbs, skull, and clavicle of E18.5 *Usp8^fl/fl^* and *Usp8^Osx^* embryos (*n* = 3/group). Scale bar, 1 mm. (**B**,**C**) Alizarin red/alcian blue staining of skeletal preparations of hindlimbs of P0 neonates (B, *n* = 3/group) and E.15.5 embryos (C, *n* = 3/group). Scale bar, 1 mm. (**D**) Safranin O–stained longitudinal sections of P0 *Usp8^fl/fl^* and *Usp8^Osx^* femurs (*n* = 3/group). Scale bar, 250 μm (left) and 100 μm (right, enlarged insert image). (**E**) Immunohistochemistry (IHC) for Bglap in the trabecular bone of P0 *Usp8^fl/fl^* and *Usp8^Osx^* femurs (*n* = 3/group). Upper: IgG control, middle and down: IHC for Bglap. Scale bar, 100 μm.

**Figure 4 ijms-22-10289-f004:**
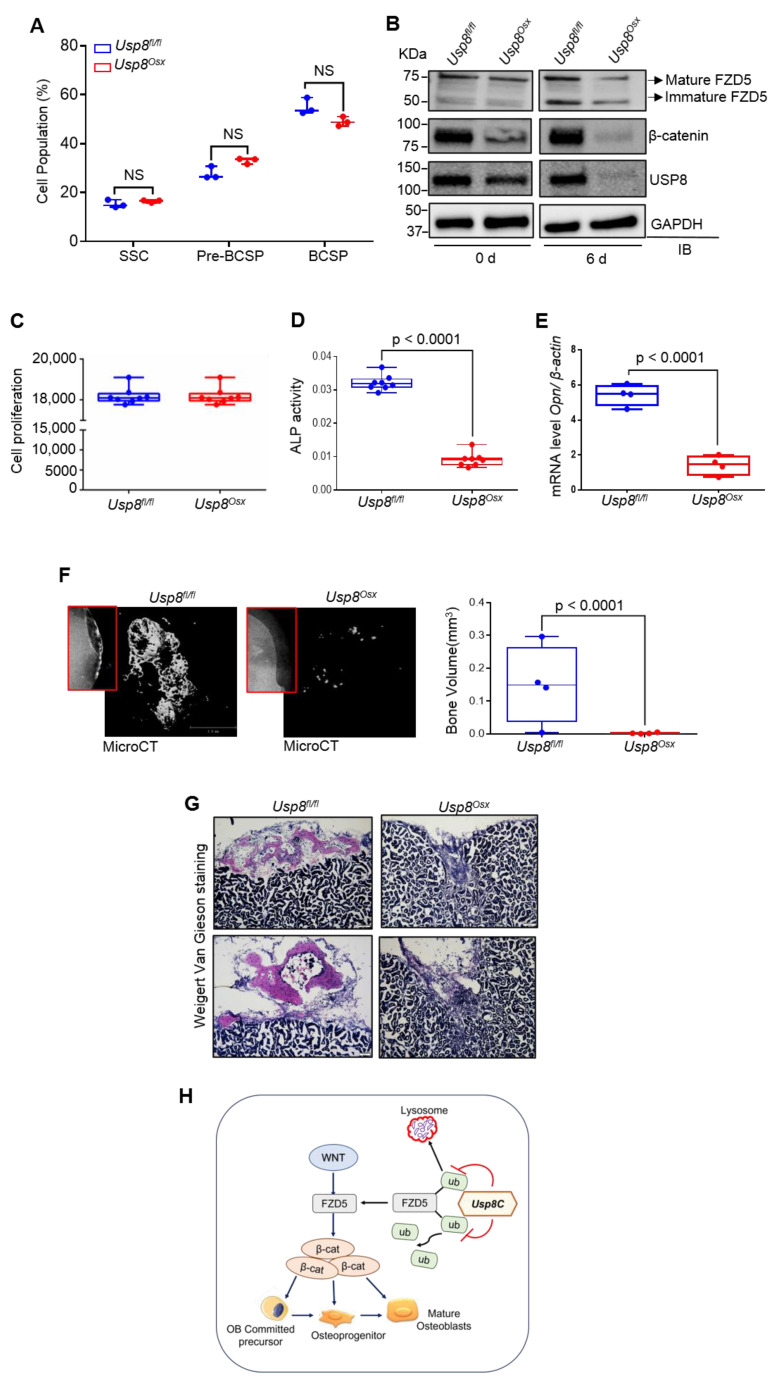
USP8 is essential for osteogenic differentiation of skeletal progenitors. (**A**) Quantification of flow cytometry on subpopulations of skeletal progenitors isolated from E18.5 *Usp8^fl/fl^* and *Usp8^Osx^* embryonic limbs (*n* = 3/group). SSC (CD45^−^Ter119^−^Tie2^−^αVInt^+^Thy^−^CD105^−^CD200^+^); pre-BCSP (CD45^−^Ter119^−^Tie2^−^αVInt^+^Thy^−^CD105^−^CD200^−^); BCSP (CD45^−^Ter^−^119^−^Tie2^−^αVInt^+^Thy^+^CD105^+^). (**B**) *Usp8^fl/fl^* and *Usp8^Osx^* SSCs (*n* = 5/group) were cultured under undifferentiated or osteogenic conditions for 6 days, and protein levels of FZD5, β–catenin, and USP8 were assessed by immunoblotting. (**C**–**E**) *Usp8^fl/fl^* and *Usp8^Osx^* SSCs (*n* = 5/group) were cultured under osteogenic conditions for 6 days, and cell proliferation (C), alkaline phosphatase activity (ALP, D) and mRNA levels of *Opn* (E) were assessed. (**F**,**G**) *Usp8^fl/fl^* and *Usp8^Osx^* SSCs (*n* = 5/group) were transplanted beneath the kidney capsule of 8–week-old male mice (*n* = 4/group), and 8 weeks later bone formation was analyzed using X–radiography (left, red box) and microCT analysis. Representative 3D–reconstruction (F, left) and relative quantification (F, right) of bone mass in the kidney capsule were displayed. Alternatively, longitudinal sections of the kidney capsule were stained with Weigert Van Gieson (G). (**H**) A schematic diagram showing molecular actions of the DUB USP8 in the regulation of the Wnt/FZD/β–catenin pathway. USP8 deubiquitinates ubiquitinylated FZD5 through the C–terminal DUB domain (USP8C) in osteoblast precursors, important for Wnt/β–catenin signaling, the differentiation of osteoprogenitors to osteoblasts. Scale bars, 200 μm. Data are shown as box–and–whisker plot (with median and interquartile ranges) from min to max, with all data points shown. Analyses were performed using the unpaired Student’s *t*–test within Prism. NS, not significant.

## Data Availability

Not applicable.

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
