# Peer review of "Deubiquitinating Enzyme USP8 Is Essential for Skeletogenesis by Regulating Wnt Signaling"

_ijms, 2021, doi:10.3390/ijms221910289_

Round 1
Reviewer 1 Report
The present manuscript addresses the implications of mice USP8 deubiquitinating enzyme knockout for osteoblasts differentiation and bone formation. The conditional knockout of the USP8 gene is used in cultured cells and animal models. On the molecular level, the Authors observe that USP8 KO cells show decreased FZD5 and catenin beta levels. Authors also observe a strong phenotype of USP8 at the animal and cell levels – differentiation into osteoblasts and bone formation are strongly decreased. Authors claim that their experiments provide evidence that USP8 mediates deubiquitination of FZD5 receptor. This, in turn, rescues FZD5 from autophagic degradation and supports Wnt signaling necessary for bone progenitors differentiation. In my opinion, these claims are not supported by the results. None of the experiments addresses the ubiquitination of FZD5. Degradation-related experiments do not allow unambiguous interpretation. The manuscript does not provide evidence for the direct role of USP8. The effect of the knockout on bone development could be indirect and related to the arrest of the cell cycle, which was previously reported for Usp8 knockout in human cells. In addition, the effect of Usp8 on FZD5 degradation was published before, limiting the present study's novelty. For these reasons, I can't recommend the manuscript for publication in the International Journal of Molecular Sciences. Below I list specific concerns:
Fig 1B,G Do numbers under the blots represent densitometry of the presented images, or do they represent the mean of several repetitions? As the differences in band intensities are not clear, these data should be analyzed for statistical significance.
Fig 1C Is the knockout complete? There are still visible bands at the size corresponding to Usp8 in the lanes that contain KO cells?
Fig 1E. I have no background in bone morphology, but femur sections look very different. Do differences represent the Usp8 phenotype or different regions of the bone were taken?
Fig 1F, Authors, claim that b-catenin activity decrease caused by Usp8 deletion was reverted by the expression of c-terminal part of USP8. However, the experiment does not include cells with normal expression of Usp8 to compare. Thus it can be only concluded that the activity of b-catenin was increased due to the expression of C-term USP8 as compared to deletion of USP8.
Fig 1 H,I The changes on the blot are small. It is difficult to judge if Bafilomycin A treatment indeed prevented FZD5 degradation as the necessary control of Usp8 deletion without Bafilomycin treatment was not included in the experiment. Also, the difference in the band intensities are minimal to the eye. To draw conclusions, quantification of repeated experiments should be presented, and statistical significance should be verified.
Also, why FZD5 was ectopically expressed in the experiment? Previous experiment (1D) showed depletion of native FZD5?
Figure 1 F, G Results For Usp8N seem contradictory?
Minor:
The text refers to non-existent Figure 4 (line 187). This should be Figure 3G
Fig 1F,G a schematic diagram showing usp8 domains and fragments used in experiments would facilitate result interpretation
Author Response
- We agree with the reviewer’s concerns, appreciate the reviewer’s constructive suggestions, and believe that addressing these has strengthened the revised manuscript.
None of the experiments addresses the ubiquitination of FZD5. Degradation-related experiments do not allow unambiguous interpretation. The manuscript does not provide evidence for the direct role of USP8.
- Using the HEK293T overexpression system, our prior work (Chen et al., eLife, 2020;9:e56793) demonstrated that 1) USP8 interacts with the WNT receptor FZD5 and 2) the deubiquitination enzyme (DUB) domain of USP8 was sufficient to inhibit the ubiquitination of FZD5 while enhancing WNT/b-catenin signaling. Consistent with these biochemical analyses, our results demonstrated that in primary osteoprogenitors, 1) USP8 interacts with FZD5 and 2) USP8-deficiency increased ubiquitination of FZD5, leading to a decrease in protein levels of FZD5 and b-catenin and WNT/b-catenin signaling activity.
The effect of the knockout on bone development could be indirect and related to the arrest of the cell cycle, which was previously reported for Usp8 knockout in human cells.
- In contrast to human cells lacking USP8 with the arrest of the cell cycle, USP8-deficient osteoprogenitors showed normal cell proliferation. Similarly, USP8-deficiency in embryonic stem cells did not affect apoptosis (Gu et al., Nat Commun., 2019;10:1465). Finally, mice lacking USP8 in osteoprogenitors (USP8osx mice) phenocopied the skeletal phenotypes seen in mice with WNT signal deficiency, including mice lacking b-catenin in osteogenic progenitors (Ye et al., JBMR, 2018;36:560, Hill et al., Dev. Cell, 2005;8:727).
The effect of Usp8 on FZD5 degradation was published before, limiting the present study's novelty.
- Although the effects of USP8 on FZD5 degradation were previously described by the biochemical analyses using the HEK293T overexpression system, this study is the first to report showing in vivo roles of USP8 in controlling WNT/b-catenin signaling in osteoprogenitors during skeletal development.
Major points:
1. None of the experiments addresses the ubiquitination of FZD5. The manuscript does not provide evidence for the direct role of USP8.
- We appreciate the reviewer for pointing this out. Our prior work identified USP8 as the bona fide deubiquitinating enzyme of FZD5 using the HEK293T overexpression system (Chen et al., eLife, 2020;9:e56793). 1) USP8 interacts directly with FZD5 and 2) the DUB domain of USP8 inhibited the ubiquitination of FZD5 while enhancing WNT/b-catenin signaling. Consistent with these biochemical analyses, our results demonstrated that in primary osteoprogenitors, 1) USP8 interacts with FZD5 (Figure 2A), 2) USP8-deficiency increased ubiquitination levels of FZD5 (Figure 2D), leading to a decrease in protein levels of FZD5 (Figure 2B) and b-catenin (Figure 1D) and WNT/b-catenin signaling activity (Figure 2E), and 3) these phenotypes were partly reversed by treatment with the lysosomal inhibitor Bafilomycin A1 (Figure 2C) or by enforced expression of the DUB domain of USP8 or FZD5 (Figure 2E-G). The revised version of this data and our new data were added to the revised manuscript.
2. The effect of the knockout on bone development could be indirect and related to the arrest of the cell cycle, which was previously reported for Usp8 knockout in human cells.
- We appreciate the reviewer for pointing this out. To test this possibility, cell proliferation assay in WT and USP8-deficient osteoprogenitors was performed using Alamar Blue staining, demonstrating little to no effect of USP8-deficiency on cell proliferation rate (Figure 4C). Similarly, USP8-deficiency in embryonic stem cells did not affect apoptosis (Gu et al., Nat Commun., 2019;10:1465). Finally, mice lacking USP8 in osteogenic progenitors (USP8osx mice) phenocopied the skeletal phenotypes seen in mice with WNT signal deficiency, including mice lacking b-catenin in osteogenic progenitors (Ye et al., JBMR, 2018;36:560, Hill et al., Dev. Cell, 2005;8:727). Thus, these results suggest that USP8 is dispensable for cell cycle regulation in osteoprogenitors and the skeletal defects seen in USP8osx mice may result from impaired WNT/b-catenin signaling, not cell cycle arrest.
3. Degradation-related experiments do not allow unambiguous interpretation. In addition, the effect of Usp8 on FZD5 degradation was published before, limiting the present study's novelty.
- We agree with the reviewer’s concern that degradation-related experiments do not allow unambiguous interpretation. This issue was fully addressed in the revised manuscript by adding new results (Figure 2). Although the effects of USP8 on FZD5 degradation were previously described by the biochemical analyses using the HEK293T overexpression system (Chen et al., eLife, 2020;9:e56793), this study is the first to report showing in vivo roles of USP8 in controlling WNT/b-catenin signaling in osteoprogenitors during the skeletal development.
4. Fig 1B,G Do numbers under the blots represent densitometry of the presented images, or do they represent the mean of several repetitions? As the differences in band intensities are not clear, these data should be analyzed for statistical significance.
- Figures represented the mean of several repetitions. As suggested by the reviewer, statistical significance was added to the revised Figures 1C and 2G.
5. Fig 1C Is the knockout complete? There are still visible bands at the size corresponding to Usp8 in the lanes that contain KO cells?
- Thank you for pointing this out. Although the knockout was complete by using homozygous allele of Usp8-floxed mice with the Osterix promoter-driven expression of Cre recombinase (Usp8fl/fl; Osx-cre), primary osteoprogenitors isolated from the calvaria of Usp8fl/fl;Osx-cre pups were heterogeneous, showing the different degree of USP8 deletion.
6. Fig 1E. I have no background in bone morphology, but femur sections look very different. Do differences represent the Usp8 phenotype or different regions of the bone were taken?
- Thank you for pointing this out. Immunohistochemistry for b-catenin was performed in E18.5 Usp8fl/fl and Usp8fl/fl;Osx-cre femurs, and b-catenin expression in the primary ossification center of these femurs was displayed in Figure 1E (indicated by the box in Supplementary Figure 1C). Of note, Usp8fl/fl;Osx-cre femurs showed a complete absence of skeletal elements in the primary ossification center despite expansion of chondrocytes.
7. Fig 1F, Authors, claim that b-catenin activity decrease caused by Usp8 deletion was reverted by the expression of c-terminal part of USP8. However, the experiment does not include cells with normal expression of Usp8 to compare. Thus, it can be only concluded that the activity of b-catenin was increased due to the expression of C-term USP8 as compared to deletion of USP8.
- As suggested by the reviewer, transcription activity and protein levels of b-catenin in WT cells were also added to the revised Figures 2E-G.
8. Fig 1 H, I The changes on the blot are small. It is difficult to judge if Bafilomycin A treatment indeed
prevented FZD5 degradation as the necessary control of Usp8 deletion without Bafilomycin treatment was not included in the experiment. Also, the difference in the band intensities are minimal to the eye. To draw conclusions, quantification of repeated experiments should be presented, and statistical significance should be verified.
- As suggested by the reviewer, these experiments were repeated along with the proper control (Usp8 deletion without Bafilomycin treatment), demonstrating a significant increase in protein levels of immature FZD5 in USP8-deficient osteoprogenitors when treated with Bafilomycin A1 (Figure 2C). This is consistent with previous reports showing that treatment with the lysosomal inhibitor Bafilomycin A1 preferentially affects the stability of immature FZD receptors that are primarily localized in the ER (Yamamoto et al., Cell, 2005;120:223, Koo et al., Nature, 2012;488:665). We regret to say that quantification of repeated experiments could be not performed due to limited time for the revision.
9. Also, why FZD5 was ectopically expressed in the experiment? Previous experiment (1D) showed depletion of native FZD5?
- Thank you for pointing this out. This experiment was repeated with the proper control, showing decent deletion of FZD5 in USP8-deficient osteoprogenitors (Figure 2C).
10. Figure 1 F, G Results For Usp8N seem contradictory?
- Thank you for pointing this out. The experiments for Figures 1F and G were repeated and the effects of enforced expression of USP8N on transcription activity and protein levels of β-catenin in USP8-deficient osteoprogenitors were consistent in the revised Figures 2E-G.
Minor points:
1. The text refers to non-existent Figure 4 (line 187). This should be Figure 3G.
- Thank you for the correction. It was changed to Figure 4H.
2. Fig 1F,G a schematic diagram showing usp8 domains and fragments used in experiments would facilitate result interpretation
- As suggested by the reviewer, the schematic diagram was revised (Figure 4H).

Reviewer 2 Report
Chaugule et al. present an interesting story by which the deubiquitinase (DUB) USP8 is important for skeletogenesis through Wnt signaling. DUBs, including USP8, play key roles in development and have been shown to regulate Wnt signaling. Since the disruption of this pathway is known to cause skeletal abnormalities, it seems reasonable that USP8 would function is in skeletal development. The authors present as their molecular mechanism that USP8 deubiquitinates FZD5, stabilizing FZD5 protein levels.
Here, the authors cleanly demonstrate that the deletion of USP8 impacts bone formation and differentiation, but the link to Wnt signaling is weakly examined. First, the effects of deleting Usp8, largely presented in figure 1F-G, are relatively minor. Second, the authors examined whether FZD destruction was proteasomal- or lysosomal-dependent using MG132 or Bafilomycin A1, respectively. Though the authors suggest Bafilomycin A1 treatments resulted in elevated levels of FZD5, this is not clearly demonstrated (Fig 1I). Controls for the compound or the USP8 deletions are not shown. If anything, the FZD5 levels remain the same or slightly lower when the inhibitor is added, but that the inhibitor was only added to the knockout cells, making the experiment largely uninterpretable. These experiments should be performed side-by-side and quantified if possible. Lastly, the specific molecular mechanism remains elusive. It is unclear if USP8 directly or indirectly associates with or binds to FZD5. Ultimately, it is my opinion that the authors need to firm up this side of the story to define a causative relationship between USP8 and Wnt signaling during skeletogenesis.
Minor points:
Section title reads “This USP8”. It doesn’t need “This” in the title.
The authors reference a fig 4, but figure 4 is not presented. Likely they are referring to Fig 3G.
Author Response
- We agree with the reviewer’s concerns, appreciate the reviewer’s constructive suggestions, and believe that addressing these has strengthened the revised manuscript.
- Our prior work identified USP8 as the bona fide deubiquitinating enzyme of the WNT receptor FZD5 using the HEK293T overexpression system (Chen et al., eLife, 2020;9:e56793). We demonstrated that 1) USP8 interacts directly with FZD5 and 2) the deubiquitination enzyme (DUB) domain of USP8 inhibited the ubiquitination of FZD5 while enhancing WNT/beta-catenin signaling. Consistent with these biochemical analyses, our results demonstrated that in primary osteoprogenitors, 1) USP8 interacts with FZD5 (Figure 2A), 2) USP8-deficiency increased ubiquitination levels of FZD5 (Figure 2D), leading to a decrease in protein levels of FZD5 (Figure 2B) and beta-catenin (Figure 1D) and WNT/beta-catenin signaling activity (Figure 2E), and 3) these phenotypes were partly reversed by treatment with the lysosomal inhibitor Bafilomycin A1 (Figure 2C) or by enforced expression of the DUB domain of USP8 or FZD5 (Figure 2E-G). These were added to the result section of the revised manuscript to strengthen the link between USP8 and WNT/beta-catenin signaling in osteoprogenitors.
First, the effects of deleting Usp8, largely presented in figure 1F-G, are relatively minor.
- We agree with the reviewer’s concern. Figures 1F-G were replaced with newly revised Figures 2C, E, F, and G, clarifying that a decrease in protein levels of beta-catenin and FZD5 and beta-catenin activity in USP8-deficient osteoprogenitors was reversed by treatment with Bafilomycin A1 or by expression of the DUB domain of USP8 or FZD5.
Second, the authors examined whether FZD destruction was proteasomal- or lysosomal-dependent using MG132 or Bafilomycin A1, respectively. Though the authors suggest Bafilomycin A1 treatments resulted in elevated levels of FZD5, this is not clearly demonstrated (Fig 1I). Controls for the compound or the USP8 deletions are not shown. If anything, the FZD5 levels remain the same or slightly lower when the inhibitor is added, but that the inhibitor was only added to the knockout cells, making the experiment largely uninterpretable. These experiments should be performed side-by-side and quantified if possible.
- As suggested by the reviewer, this experiment was repeated side-by-side with the proper control (USP8 deletion without Bafilomycin A1 treatment), demonstrating a significant increase in protein levels of immature FZD5 in USP8-deficient osteoprogenitors when treated with Bafilomycin A1 (Figure 2C). This is consistent with previous reports showing that treatment with the lysosomal inhibitor Bafilomycin A1 preferentially affects the stability of immature FZD receptors that are primarily localized in the ER (Yamamoto et al., Cell, 2005;120:223, Koo et al., Nature, 2012;488:665). We regret to say that quantification of repeated experiments could be not performed due to limited time for the revision.
Lastly, the specific molecular mechanism remains elusive. It is unclear if USP8 directly or indirectly associates with or binds to FZD5. Ultimately, it is my opinion that the authors need to firm up this side of the story to define a causative relationship between USP8 and Wnt signaling during skeletogenesis.
- Thank you for pointing this out. Accompanied with our previous biochemical analyses showing a direct interaction between USP8 and FZD5 (Chen et al., eLife, 2020;9:e56793), the newly revised Figure 2A demonstrated endogenous interaction between USP8 and FZD5 in osteoprogenitors. Additionally, USP8-deficiency in osteoprogenitors increased ubiquitination levels of FZD5 (Figure 2D), leading to a decrease in protein levels of FZD5 (Figure 2B) and beta-catenin (Figure 1D) and WNT/beta-catenin signaling activity (Figure 2E). These phenotypes were partly reversed by treatment with the lysosomal inhibitor Bafilomycin A1 (Figure 2C) or by expression of the DUB domain of USP8 or FZD5 (Figure 2E-G). This is consistent with our prior work showing that the DUB domain of USP8 inhibited the ubiquitination of FZD5 while enhancing WNT/beta-catenin signaling in HEK293T cells (Chen et al., eLife, 2020;9:e56793). Thus, USP8 plays a critical role in the regulation of WNT/beta-catenin signaling in osteoprogenitors by deubiquitinating the WNT receptor FZD5, which is required for early skeletogenesis.
Minor points:
Section title reads “This USP8”. It doesn’t need “This” in the title.
- As suggested by the reviewer, it was corrected.
The authors reference a fig 4, but figure 4 is not presented. Likely they are referring to Fig 3G.
- Thank you for the correction. It was changed to Figure 4H.
Round 2
Reviewer 1 Report
The changes introduced by the authors addressed most of my concerns. New data was added that support the Authors' claims. Only the copurification presented in Fig 2 A is not convincing and can be misleading (the experiment aimed to purify FZD5 which can't be confirmed due to overlap with strong IgG signal).
In general, I'm supporting the publication of the study.
Minor:
line 366 - I think only anti-FZD5 but not anti-USP8 antibodies were used for immunoprecipitation
Author Response
The changes introduced by the authors addressed most of my concerns. New data was added that support the Authors' claims. Only the copurification presented in Fig 2 A is not convincing and can be misleading (the experiment aimed to purify FZD5 which can't be confirmed due to overlap with strong IgG signal).
- We agree with the reviewer’s concern about Fig.2A. Co-immunoprecipitation analysis with anti-FZD5 antibody was repeated and this new data replaced the original one in the revised manuscript. Since FZD5 is a 55 kDa protein and the immunoprecipitate was overlapped with the strong IgG band, rabbit trueblot anti-rabbit IgG HRP kit (Rockland, #18-8816-33) was used to remove the strong IgG band from the FZD5 band, but immunoblotting with anti-FZD5 antibody did not work under this condition. Additionally, our attempt to perform co-immunoprecipitation analysis with anti-USP8 antibody and immunoblotting with anti-FZD5 antibody using this kit was not successful.
Minor: line 366 - I think only anti-FZD5 but not anti-USP8 antibodies were used for immunoprecipitation
- Thank you for the correction. It was changed in the revised manuscript.
Reviewer 2 Report
Overall, the manuscript seems improved and presented more clearly. However, the endogenous IP to validate the direct interaction between USP8 and FZD5 is not very robust and should be improved if possible. Otherwise, the direct association is largely based on the prior publication in HEK293T cells.
Author Response
Overall, the manuscript seems improved and presented more clearly. However, the endogenous IP to validate the direct interaction between USP8 and FZD5 is not very robust and should be improved if possible. Otherwise, the direct association is largely based on the prior publication in HEK293T cells.
- We agree with the reviewer’s concern about Fig.2A. Co-immunoprecipitation analysis with anti-FZD5 antibody was repeated and this new data replaced the original one in the revised manuscript.